# Tara (*Caesalpinia spinosa*) in Natural and Agroforestry Systems under an Altitudinal Gradient in the Peruvian Andes: Responses to Soil and Climate Variation

Hipolito Murga-Orrillo [1,*], Carlos Abanto-Rodriguez [2], Luiz Fernandes Silva Dionisio [3,4], Fred William Chu-Koo [1], Gustavo Schwartz [5], Ever Nuñez Bustamante [6], Paul Michael Stewart [7], Ricardo Santos Silva Amorim [8], George Louis Vourlitis [9], Francisco De Almeida Lobo [10] and Ricardo Manuel Bardales-Lozano [11]

1.  Facultad de Zootecnia, Agronomía, Ciencias Biológicas y Acuicultura, Universidad Nacional Autónoma de Alto Amazonas (UNAAA), Yurimaguas 16501, Peru
2.  Instituto de Investigaciones de la Amazonía Peruana (IIAP), Pucallpa 25001, Peru
3.  Universidade Estadual da Região Tocantina do Maranhão, Imperatriz 65910-100, Brazil
4.  Fundação Amazônia de Amparo a Estudos e Pesquisas, Belém 66060-575, Brazil
5.  Embrapa Amazônia Oriental, Manaus 69010-970, Brazil
6.  Universidad Nacional Autónoma de Chota (UNACH), Chota 06120, Peru
7.  Department of Biological and Environmental Sciences, Troy University, Troy, AL 36079, USA
8.  Universidade Federal de Viçosa (UFV), Viçosa 36570-900, Brazil
9.  Biological Sciences Department, California State University, San Marcos, CA 92024, USA
10. Universidade Federal de Mato Grosso (UFMT), Cuiabá 78060-900, Brazil
11. Facultad de Agronomía, Universidad Nacional de la Amazonía Peruana (UNAP), Iquitos 16001, Peru
*   Correspondence: hmurga@unaaa.edu.pe

**Abstract:** This research examined how edaphoclimatic variations are related to dendrometric variables of the Tara tree in natural and agroforestry systems in Cajamarca, Peru. Evaluations followed three approaches: (a) principal components analysis (PCA) with environmental factors and altitude classes with 1 °C of temperature variation for edaphic and dendrometric variables; (b) evaluation of possible differences by the bootstrap method for the different variables in the PCAs; and (c) correlation analysis between plant density, plant and crown height, stem and crown diameter, and the number of stem branches with the physical and chemical attributes of the soil and with air temperature. In the altitudinal gradient from 2021 to 3007 ± 7 m, the temperature ranged from 19.8 to 13.4 ± 0.4 °C; the soils possessed alkaline pH, high organic matter (OM), K and CEC, lower contents of CaCO₃, N, P, K, B, Cu, Fe, Mn, and Zn. The soil properties with the most significant contribution to PCAs were OM, CEC, N content, and sand, with no variation among environments but among altitudes. None of the dendrometric variables varied as a function of altitude and temperature in PCAs run in the natural environment. However, in the agroforestry environment, there was a greater crown diameter, and tree and crown height in the 2185 m altitude class associated with *Medicago sativa*. In contrast, the opposite behavior was found in these variables and in the altitude class 2798 m associated with low Fe content, and already in the altitude class at 3007 m, a larger stem diameter is associated with higher levels of CaCO₃. The anthropic effect on the agroforestry environment did not significantly alter the soil's CEC, OM, N, and sand. Tara's crown diameter and tree height appeared higher in agroforestry environments. The variations of Fe and CaCO₃ in the soil might have influence on the development of Tara individuals in natural and agroforestry environments. It is important to carry out further studies for a better understanding of the relationship between the production of Tara pods and soil fertility in altitudinal variation, aiming to improve the income and employment of family farmers who exploit Tara in the Peruvian Andes.

**Keywords:** soil; temperature; altitude; tropical mountains; dendrometric

## 1. Introduction

The intensification of agriculture and the exploitation of natural resources have generated significant changes in tropical mountain forests [1], affecting many populations of tree species in this environment. The Tara, *Caesalpinia spinosa* (Molina) Kuntze, is a species of the Fabaceae (Caesalpiniaceae) family endemic to the South American tropical Andes [2]. In the region of Cajamarca, Peru, Tara is subject to altitude gradients, which generate constant variations in air and soil temperature.

Over the last few decades, Tara has been widely used as a source of firewood and has also suffered from the expansion of agriculture and livestock, which has resulted in a significant reduction of its natural populations. Currently, the species is found only in a few natural forest remnants or integrated into agroforestry systems. There has been an increase in its economic value due to the multiple uses of its pods as a source of raw material in various industries. The pods contain tannins with anticancer, antioxidant, antimicrobial, and anticorrosive activity, among others [3,4]. Seed gum is used as a biopolymer or biocoagulant in water treatment [5,6] and in the food and cosmetics industries [7,8].

In both natural remnants and agroforestry systems, Tara has low natural regeneration, which increases the need to conserve the species [9]. Tara forests contribute to soil conservation in the sloping areas of the Andes watershed [10]. Agroforestry systems improve agronomic productivity, carbon stock, nutrient cycling, soil biodiversity, water retention, and pollination, and decrease soil erosion and the incidence of fires, thus providing recreational and cultural benefits [11]. In the Andes of Cajamarca, Tara individuals in agroforestry systems are found within and on the edge of small agricultural areas, such as hedges and windbreaks, which helps reduce erosion risks and gullies, stabilize steep soil slopes, and generate shade.

The remaining Tara forests are distributed in altitudinal gradients of the South American Andes, primarily in the north of Peru in areas with complex topography in the tropical mountain forest. Between 1000 and 3500 m altitude, tropical cloud mountain forests are found [12]. These forests are unique among terrestrial ecosystems and are strongly linked to the regulatory cycles of cloud formation [13–15].

According to Holdridge [16] and Malizia et al. [17], mountain forests are classified in consonance with altitudinal variation into pre-mountain (1000 to 2000 m), low mountain (2000 to 3000 m), and mountain (3000 to 4000 m) categories. In these altitudinal variations, a thermal variation of $-0.48$ to $-0.65$ °C is observed for each 100 m increase in altitude [18,19]. Thermic variation influences soil processes, thus generating environmental filters that influence ecological and biogeographic processes of species on a local and temporal scale [20–22].

In addition to altitudinal variation, the mountain environment possesses soils with different physical and chemical attributes than lowland areas, which may vary tree species' dendrometry [14,22,23]. As soil management is less intensive in agroforestry systems, soil properties in these environments are similar to those of natural systems [24]. In natural and agroforestry systems, tree roots and fauna produce aggregates that maintain soil properties in these environments [25].

Studies on the effects of soil and climate factors on natural remnants of Tara in tropical mountain forests are scarce. Thus, knowledge regarding these conditions will hopefully lead to the conservation and improvement of this species' natural and agroforestry management. It is thought that the natural and agroforestry environments and the altitudinal gradient that exists influences soil properties and the allometry of Tara trees; therefore, we intend to answer the following question: How does the altitudinal gradient, with its influence on edaphic and climatic properties, affect Tara dendrometric variables in the Peruvian Andes?

## 2. Material and Methods

### 2.1. Characterization of the Study Area

The study area covered the province of San Marcos, region of Cajamarca, Peru (Figure 1), in the Andes, between 7.2 and 7.6° S latitude, 79.9 and 78.4° W. longitude, in an altitudinal gradient of 2000 to 3200 m. We sampled in areas of the "Cooperativa Agraria de Productores de Tara del Norte—APT del Norte", which contains 120 family farming partners, who exploit Tara under natural conditions and agroforestry systems for production and commercialization.

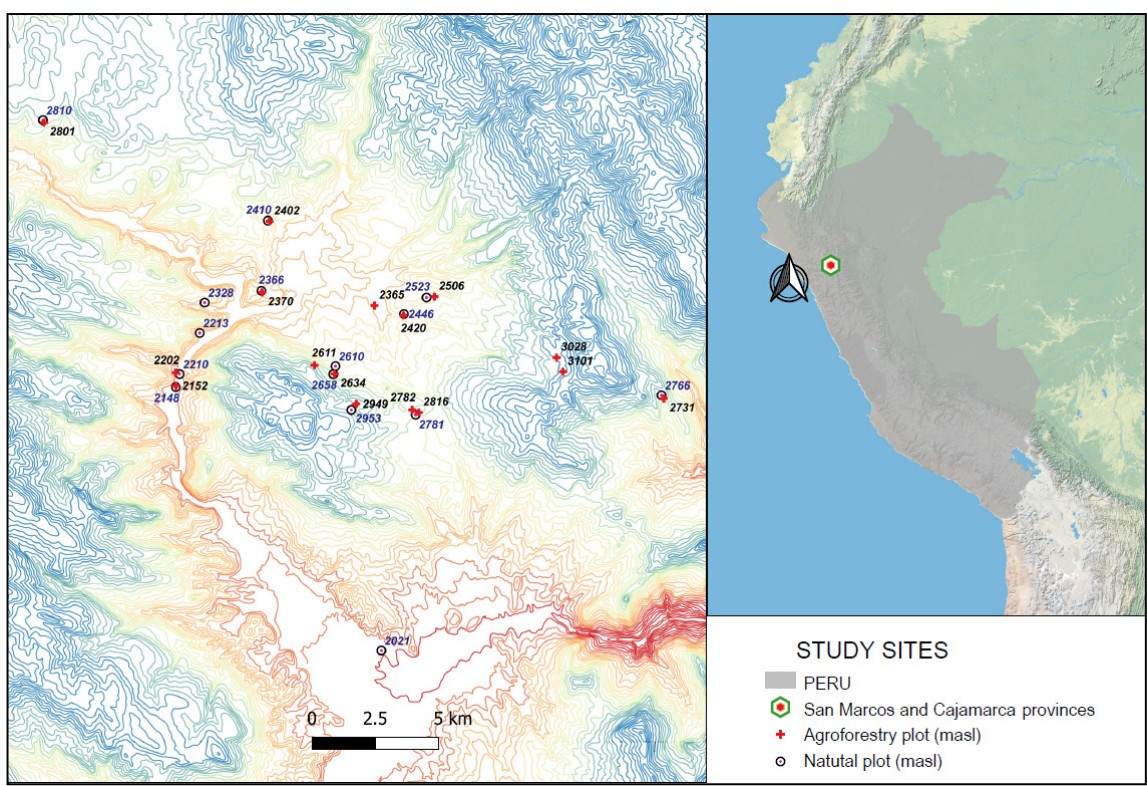

**Figure 1.** Location of the study area and distribution of plots in the region of Cajamarca, Andes, Peru.

The study area has a dry season between April and September and a rainy season between October and March. According to the classification by Holdridge [16], the climate of the study area is tropical, with low mountains and mountain forests. The mean annual temperature (MAT) and the mean annual precipitation (MAP) in low mountains vary by 12–18 °C and 6000–8000 mm, and in the mountains vary by 6–12 °C and 4000–6000 mm, respectively [26].

### 2.2. Delimitation of Installments

Thirty-one plots were distributed, from low altitude to high, thus forming an altitudinal gradient (Figure 1). The plots were grouped into two sets: 15 plots were established in an area with natural forests, between 2021 and 2953 m altitude. A second set was established in agroforestry systems associated with Tara, with 16 plots between 2152 and 3101 m altitude. Of the plots in agroforestry systems, three were associated with *Medicago sativa* and *Lolium multiflorum*; eight plots were associated with *Zea mays* and *Phaseolus vulgaris*; two were associated with *Triticum aestivum*; one was associated with *Linum usitatissimum*; and two associated with *Z. mays* and *Vicia faba*.

The plots, with an area of 200 m² (20 m × 10 m) and length parallel to the contours of the soil topography, were georeferenced by a central point with the aid of a mobile phone (Samsung Galaxy J5 Prime) and the CR Campeiro 7 software (Universidade Federal de Santa Maria, Rio Grande do Sul, Brasil).

### 2.3. Soil Sampling and Analysis

Five 1 kg soil subsamples were collected in each plot, at a depth of 0 to 40 cm, which were later mixed, forming a composite sample. Each subsample mixture weighed a total of 5 kg, which was homogenized in a single sample of 1 kg of soil per plot, following the methodology of Embrapa [27].

The samples were air-dried at an ambient temperature and then refrigerated at a temperature of 0 °C, for further analysis. Analyses were performed in the "Laboratorio de Análisis de Suelos, Plantas, Aguas y Fertilizantes of the Universidad Nacional Agraria La Molina". The following soil attributes were measured: texture, pH, electrical conductivity (EC), phosphorus (P), nitrogen (N), calcium (Ca), magnesium (Mg), sodium (Na), potassium (K), iron (Fe), copper (Cu), zinc (Zn), manganese (Mn), boron (B), organic matter (OM), calcium carbonate ($CaCO_3$), and cation exchange capacity (CEC). All measurements were taken according to standard methodology. Texture was analyzed by using the Bouyoucos hydrometer. The pH measurement was performed using a potentiometer in water 1:2.5; EC was performed using the aqueous extract 1:1.5 conductivity meter; for the determination of P, Olsen modified method was used in extraction with $NaHCO_3$ 0.5 N, pH 8.5; for N, Kjeldahl method; for Ca, Mg, Na, and K, extraction in 1.0 N $NH_4Ac$, pH = 7, and atomic absorption spectrophotometry reading; for Fe, Cu, Zn, and Mn, extraction in Hunter solution and atomic absorption spectrophotometry reading were used. The CEC was obtained by the sum of $Ca^{2+}$, $Mg^{2+}$, $K^+$, $Na^+$, $Al^{3+}$, and H+ cations, obtained by Schahtschabel $NH_4Ac$ 1.0 N and atomic absorption spectrophotometry reading; B by colorimetric curcumin in glacial acetic acid; OM by Walkley and Black modified by oxidation on 4.0 N $Na_2Cr_2O_7$ + 10.0 N $H_2SO_4$; and $CaCO_3$ by gas volumetric. Embrapa's [27] classification was used for the soil's chemical properties.

### 2.4. Precipitation and Temperature Data

For precipitation, 30-year historical data (1989 to 2018) from three weather stations were used (San Marcos, Cajabamba and Huamachuco) (Table 1). For temperature, 25-year data (1989 to 2014) of the monthly average of nine meteorological stations near the study area were used (Table 1).

**Table 1.** Weather stations, with latitude ($\phi$, °S), longitude ($\lambda$, °W), and altitude ($z$, masl) for 25-year historical data.

| Station | $\phi$ | $\lambda$ | $z$ | Station | $\phi$ | $\lambda$ | $z$ |
|---|---|---|---|---|---|---|---|
| San Marcos | 7.3 | 78.2 | 2293 | Chancaybaños | 6.6 | 78.9 | 1639 |
| Cajabamba | 7.6 | 78.1 | 2626 | Cutervo | 6.4 | 78.8 | 2622 |
| Huamachuco | 7.8 | 78.0 | 3186 | Cochabamba | 6.5 | 78.9 | 1653 |
| Augusto Weberbauer | 7.2 | 78.5 | 2673 | Llama | 6.5 | 79.1 | 2096 |
| Asunción | 7.3 | 78.5 | 2270 | - | - | - | - |

The meteorological data were made available by SENAMHI [28] and used to generate a multiple linear regression model of the monthly temperature (Equation (1)) as a function of longitude ($\lambda$) and altitude ($z$), and astronomical length of the day ($\Omega$). Significant parameters of $p < 0.001$ were obtained for $\lambda$, $\Omega$ and $\Omega * \lambda$, of $p < 0.01$ for $z$, $\lambda * z * \Omega$, with $R^2$ Aj = 0.89. Equation (2) (Equation (2)) was used to estimate the $\Omega$ while determining the hourly angle of sunrise or sunset ($H$) and the declination angle of the sun ($\delta$) for the meteorological stations (Equations (3) and (4)). This model estimated the monthly temperatures characteristic of each plot studied in order to determine the MAT.

$$MMT = -4175.747 - 53.16399\lambda + 1.029125z + 375.7518\Omega + 0.01303798\lambda z$$
$$+ 4.751127\lambda\Omega - 0.08902829z\Omega - 0.001121538\lambda z\Omega \qquad (1)$$

$$\Omega = 2(H + 0.83)/15 \qquad (2)$$

$$H = arccos(-tan\phi tan\delta) \tag{3}$$

$$\delta = 23.45 sen[2\pi/365(284 + dj)] \tag{4}$$

where: *MMT*—mean monthly temperature (°C), $\lambda$—longitude (degrees), $z$—altitude (masl), $\Omega$—astronomical length of the day (h day$^{-1}$), $H$—the hourly angle of sunrise or sunset (degrees), $\phi$—latitude (degrees), $\delta$—declination angle of the sun (degrees), and *dj*—Julian days.

### 2.5. Dendrometric Variables of Tara Individuals

During the period from September 2019 to February 2020, in the 31 plots, dendrometric variables, such as total height and stem height, and average diameters of crowns and stems, of a total of 287 original trees were evaluated, corresponding to 226 natural remnants and 61 integrated into agroforestry systems with management age greater than 15 years. These evaluations were made with a measuring tape fixed to a straight wooden rod. The diameter of the single stem or multiple stems of the Tara trees was measured with a diametric tape at the height of 40 cm above the soil surface.

### 2.6. Grouping of Variables

For analysis, data were grouped by environments: (1) natural (15 plots) and (2) agroforestry (16 plots). Seven classes of altitudes were determined (2021, 2185, 2388, 2546, 2680, 2798, and 3007 m), consequently corresponding to seven classes of temperatures (19.8; 18.5; 17.3; 16.4; 15.6; 14.6, and 13.4 °C) related to the increase in altitude (Table 2, Figure S1).

**Table 2.** Geographic location and soil physical properties of plots in altitude classes, in all environments, in natural and agroforestry environments of Tara, Cajamarca region, Andes Mountain range, Peru.

| Altitude * | MAT | Latitude | Longitude | Clay | Silt | Sand | Texture |
|---|---|---|---|---|---|---|---|
| m | °C | | | | % | | |
| *Natural–agroforestry environments* | | | | | | | |
| 2021 (n = 1) | 19.8 | 7.5 | 78.2 | 34.0 | 21.0 | 45.0 | LCS |
| 2185 ± 3 (n = 4) | 18.5 ± 0.2 | 7.4 ± 0.0 | 78.2 ± 0.0 | 26.8 ± 2.3 | 33.8 ± 6.4 | 39.4 ± 7.1 | L |
| 2388 ± 4 (n = 8) | 17.3 ± 0.20 | 7.3 ± 0.0 | 78.2 ± 0.0 | 27.3 ± 7.6 | 21.3 ± 4.5 | 51.5 ± 10.5 | LCS |
| 2546 ± 6 (n = 3) | 16.4 ± 0.4 | 7.4 ± 0.0 | 78.1 ± 0.0 | 29.3 ± 10.3 | 22.3 ± 3.1 | 48.3 ± 13.3 | LCS |
| 2680 ± 7 (n = 5) | 15.6 ± 0.3 | 7.4 ± 0.0 | 78.1 ± 0.1 | 29.2 ± 5.0 | 25.8 ± 1.1 | 45.0 ± 5.5 | LCS |
| 2798 ± 2 (n = 5) | 14.6 ± 0.3 | 7.3 ± 0.1 | 78.2 ± 0.0 | 33.2 ± 13.0 | 26.2 ± 3.4 | 40.6 ± 13.5 | LC |
| 3007 ± 7 (n = 4) | 13.4 ± 0.4 | 7.4 ± 0.0 | 78.1 ± 0.0 | 28.0 ± 2.8 | 28.5 ± 6.2 | 43.5 ± 5.0 | LC |
| *Natural environment* | | | | | | | |
| 2021 (n = 1) | 19.8 | 7.5 | 78.2 | 34.0 | 21.0 | 45.0 | LCS |
| 2185 ± 4 (n = 3) | 18.5 ± 0.2 | 7.4 ± 0.0 | 78.2 ± 0.0 | 25.3 ± 1.2 | 31.7 ± 1.2 | 43.0 ± 2.0 | L |
| 2388 ± 5 (n = 4) | 17.3 ± 0.3 | 7.3 ± 0.0 | 78.2 ± 0.0 | 27.5 ± 8.1 | 20.5 ± 5.3 | 52.0 ± 12.9 | LCS |
| 2546 ± 6 (n = 2) | 16.2 ± 0.5 | 7.4 ± 0.0 | 78.2 ± 0.0 | 28.0 ± 14.1 | 22.0 ± 4.2 | 50.0 ± 18.4 | LCS |
| 2680 ± 8 (n = 2) | 15.4 ± 0.3 | 7.4 ± 0.0 | 78.1 ± 0.1 | 25.0 ± 4.2 | 25.0 ± 0.0 | 50.0 ± 4.2 | LCS |
| 2798 ± 2 (n = 2) | 14.6 ± 0.4 | 7.3 ± 0.1 | 78.2 ± 0.1 | 31.0 ± 24.0 | 23.0 ± 0.0 | 46.0 ± 24.0 | LCS |
| 3007 (n = 1) | 13.7 | 7.4 | 78.2 | 26.0 | 29.0 | 45.0 | L |
| *Agroforestry environment* | | | | | | | |
| 2185 ± 4 (n = 2) | 18.5 ± 0.2 | 7.4 ± 0.0 | 78.2 ± 0.0 | 29.0 ± 1.4 | 37.0 ± 11.3 | 34.0 ± 9.9 | LC |
| *M. sativa, L. multiflorum* | | | | | | | |
| 2388 ± 3 (n = 4) | 17.3 ± 0.2 | 7.3 ± 0.0 | 78.2 ± 0.0 | 27.0 ± 8.3 | 22.0 ± 4.2 | 51.0 ± 9.4 | LCS |
| *Z. mays, P. vulgaris, T. aestivum* | | | | | | | |
| 2546 (n = 1) | 16.7 | 7.3 | 78.1 | 32.0 | 23.0 | 45.0 | LCS |
| *Z. mays, P. vulgaris* | | | | | | | |
| 2680 ± 6 (n = 3) | 15.7 ± 0.2 | 7.4 ± 0.0 | 78.1 ± 0.1 | 32.0 ± 3.5 | 26.3 ± 1.2 | 41.7 ± 3.1 | LC |
| *Z. mays, P. vulgaris, M. sativa, L. multiflorum* | | | | | | | |
| 2798 ± 2 (n = 3) | 14.6 ± 0.3 | 7.4 ± 0.1 | 78.2 ± 0.1 | 34.7 ± 6.4 | 28.3 ± 2.3 | 37.0 ± 5.3 | LC |
| *Z. mays, P. vulgaris* | | | | | | | |
| 3007 ± 8 (n = 3) | 13.4 ± 0.4 | 7.4 ± 0.0 | 78.1 ± 0.0 | 28.7 ± 3.1 | 28.3 ± 7.6 | 43.0 ± 6.0 | LC |
| *Z. mays, V. faba, L. usitatissimum* | | | | | | | |

MAT—mean annual temperature, LCS—loam clay sandy, L—loam, LC—loam clay, n- experimental plot, *—standard deviation (±).

### 2.7. Statistical Analysis

The annual temperature estimate was performed using multiple linear regression, with quantitative analysis of data normality, residual outliers, residual independence, homoscedasticity of variances, multicollinearity, and analysis of variance of the model, based on historical information from temperature, longitude, altitude, and astronomical length of the day, as described in the item "Precipitation and temperature data".

To assess the existence of discrimination between environments by production system and by altitude classes, principal component analysis (PCA) was used. In the selection of more correlated variables (with an explanation above 70% of the total accumulated variation), the methodology of Jolliffe [29] was used. In the variables that contributed to the PCAs, 100,000 data resamplings were generated to determine the mean and non-parametric confidence intervals (CI: 95%) by the bootstrap method. In the natural and agroforestry environments, correlations were made between the average annual temperature, altitude, the number of stems, crown diameter, tree height, crown height, stem diameter, and soil to determine Pearson's correlation coefficient ($p < 0.05$). All analyses were performed using FactoMineR, factoextra, corrplot, ggplot2, tidyverse, and boot packages [30].

## 3. Results

### 3.1. Temperature and Rainfall

Individuals of Tara were found in natural and agroforestry environments, in a gradient with an altitude variation of 1062 m (2021 and $3007 \pm 7$ m) in low mountain forests mostly (<3000 m), and in tropical mountains in low quantity (>3000 m) (Table 3). In this gradient, the MAT decreased by 0.64 °C for every 100 m increase in altitude (Figure S1, Table 2); and the MAP was 770.4, 1012.7, and 1052 mm for altitudes of 2293, 2626, and 3186 m, respectively, indicating the annual water availability for Tara in this study area.

**Table 3.** Soil chemical properties of plots in altitude classes, in all environments, in natural and agroforestry environments of Tara, Cajamarca region, Andes Mountain range, Peru.

| Altitude * | CE | CaCO$_3$ | pH | OM | N | P | K | B | Cu | Fe | Mn | Zn | CEC |
|---|---|---|---|---|---|---|---|---|---|---|---|---|---|
| m | dS.m$^{-1}$ | % | | % | | | | | mg.dm$^{-3}$ | | | | cmolc.kg$^{-1}$ |
| Natural-agroforestry environment | | | | | | | | | | | | | |
| 2021 (n = 1) | 0.39 | 3.1 | 7.7 | 4.2 | 0.28 | 24.9 | 452.0 | 1.35 | 0.80 | 47.7 | 0.80 | 4.6 | 15.0 |
| 2185 ± 3 (n = 4) | 0.32 ± 0.09 | 43.8 ± 12.3 | 8.0 ± 0.3 | 5.1 ± 3.1 | 0.29 ± 0.15 | 10.7 ± 5.0 | 241.2 ± 212.4 | 0.59 ± 0.24 | 0.30 ± 0.15 | 31.7 ± 15.1 | 2.40 ± 1.93 | 4.0 ± 0.7 | 18.0 ± 7.5 |
| 2388 ± 4 (n = 8) | 0.31 ± 0.16 | 10.1 ± 12.4 | 7.8 ± 0.2 | 3.3 ± 1.5 | 0.22 ± 0.06 | 9.9 ± 5.4 | 366.1 ± 237.7 | 0.78 ± 0.46 | 0.36 ± 0.18 | 31.5 ± 22.2 | 0.42 ± 0.31 | 3.7 ± 0.4 | 19.3 ± 4.4 |
| 2546 ± 6 (n = 3) | 0.29 ± 0.09 | 18.6 ± 15.5 | 7.9 ± 0.0 | 2.8 ± 2.0 | 0.22 ± 0.16 | 8.2 ± 2.9 | 387.7 ± 272.1 | 0.44 ± 0.34 | 0.32 ± 0.35 | 27.3 ± 21.0 | 1.79 ± 2.13 | 3.9 ± 0.5 | 19.4 ± 10.4 |
| 2680 ± 7 (n = 5) | 0.30 ± 0.04 | 21.3 ± 20.3 | 7.7 ± 0.2 | 5.0 ± 2.0 | 0.33 ± 0.12 | 10.4 ± 6.0 | 452.4 ± 83.7 | 0.77 ± 0.26 | 0.40 ± 0.20 | 47.6 ± 58.0 | 0.46 ± 0.48 | 4.0 ± 0.5 | 21.1 ± 5.1 |
| 2798 ± 2 (n = 5) | 0.27 ± 0.13 | 42.9 ± 14.4 | 7.9 ± 0.1 | 5.2 ± 3.4 | 0.39 ± 0.22 | 19.3 ± 13.9 | 226.2 ± 108.5 | 0.73 ± 0.25 | 0.55 ± 0.34 | 7.0 ± 0.8 | 0.40 ± 0.58 | 3.54 ± 0.22 | 25.7 ± 8.5 |
| 3007 ± 7 (n = 4) | 0.31 ± 0.10 | 35.2 ± 12.3 | 7.9 ± 0.0 | 4.5 ± 2.6 | 0.33 ± 0.13 | 21.7 ± 12.2 | 280.3 ± 203.4 | 0.97 ± 0.76 | 0.52 ± 0.30 | 45.1 ± 54.7 | 1.20 ± 1.00 | 3.5 ± 0.4 | 23.5 ± 9.8 |
| Natural environment | | | | | | | | | | | | | |
| 2021 (n = 1) | 0.39 | 3.10 | 7.72 | 4.2 | 0.28 | 24.9 | 452.0 | 1.35 | 0.80 | 47.7 | 0.80 | 4.6 | 15.0 |
| 2185 ± 4 (n = 3) | 0.31 ± 0.11 | 47.4 ± 14.5 | 8.0 ± 0.4 | 5.2 ± 4.0 | 0.32 ± 0.18 | 12.6 ± 6.0 | 303.7 ± 271.8 | 0.69 ± 0.29 | 0.21 ± 0.05 | 36.5 ± 9.9 | 3.73 ± 0.88 | 4.4 ± 0.7 | 17.2 ± 8.8 |
| 2388 ± 5 (n = 4) | 0.23 ± 0.11 | 14.9 ± 16.9 | 7.9 ± 0.2 | 3.5 ± 1.5 | 0.23 ± 0.06 | 8.4 ± 1.6 | 315.3 ± 125.1 | 0.89 ± 0.28 | 0.44 ± 0.21 | 17.7 ± 9.9 | 0.26 ± 0.21 | 3.7 ± 0.4 | 17.6 ± 4.14 |
| 2546 ± 6 (n = 2) | 0.25 ± 0.07 | 18.8 ± 21.9 | 7.9 ± 0.0 | 3.3 ± 2.6 | 0.27 ± 0.19 | 9.6 ± 2.5 | 434.0 ± 367.7 | 0.59 ± 0.30 | 0.12 ± 0.06 | 15.3 ± 3.4 | 2.44 ± 2.55 | 3.9 ± 0.8 | 20.7 ± 14.4 |
| 2680 ± 8 (n = 2) | 0.30 ± 0.06 | 17.5 ± 24.5 | 7.8 ± 0.0 | 4.4 ± 3.5 | 0.25 ± 0.14 | 7.6 ± 0.3 | 425.5 ± 99.7 | 0.92 ± 0.30 | 0.44 ± 0.17 | 20.7 ± 4.1 | 0.16 ± 0.00 | 3.2 ± 0.3 | 17.3 ± 5.0 |
| 2798 ± 2 (n = 2) | 0.29 ± 0.20 | 42.2 ± 13.2 | 7.9 ± 0.2 | 6.1 ± 4.9 | 0.46 ± 0.34 | 12.2 ± 8.7 | 233.0 ± 91.9 | 0.71 ± 0.30 | 0.68 ± 0.28 | 6.9 ± 0.4 | 0.12 ± 0.06 | 3.6 ± 0.4 | 24.8 ± 9.7 |
| 3007 (n = 1) | 0.24 | 18.10 | 7.83 | 4.6 | 0.40 | 7.5 | 139.0 | 0.16 | 0.56 | 125.3 | 1.36 | 3.5 | 36.5 |
| Agroforestry environment | | | | | | | | | | | | | |
| 2185 ± 4 (n = 2) | 0.34 ± 0.07 | 38.4 ± 9.1 | 7.9 ± 0.4 | 5.0 ± 2.7 | 0.26 ± 0.12 | 7.9 ± 1.7 | 147.5 ± 58.7 | 0.45 ± 0.08 | 0.44 ± 0.17 | 24.5 ± 23.2 | 0.40 ± 0.23 | 3.5 ± 0.1 | 19.2 ± 8.2 |
| *M. sativa, L. multiflorum* | | | | | | | | | | | | | |
| 2388 ± 3 (n = 4) | 0.39 ± 0.16 | 5.4 ± 3.4 | 7.7 ± 0.1 | 3.1 ± 1.6 | 0.22 ± 0.07 | 11.4 ± 7.8 | 417.0 ± 330.6 | 0.67 ± 0.63 | 0.28 ± 0.10 | 45.1 ± 23.5 | 0.58 ± 0.34 | 3.8 ± 0.4 | 21.0 ± 4.5 |
| *Z. mays, P. vulgaris, T. aestivum* | | | | | | | | | | | | | |
| 2546 (n = 1) | 0.37 | 18.10 | 7.92 | 1.7 | 0.12 | 5.5 | 295.0 | 0.14 | 0.72 | 51.4 | 0.48 | 3.8 | 16.8 |
| *Z. mays, P. vulgaris* | | | | | | | | | | | | | |
| 2680 ± 6 (n = 3) | 0.30 ± 0.03 | 23.9 ± 22.3 | 7.7 ± 0.3 | 5.4 ± 1.1 | 0.38 ± 0.08 | 12.2 ± 7.7 | 470.3 ± 88.5 | 0.67 ± 0.24 | 0.37 ± 0.24 | 65.5 ± 74.2 | 0.67 ± 0.46 | 3.8 ± 0.7 | 23.7 ± 3.9 |
| *Z. mays, P. vulgaris, M. sativa, L. multiflorum* | | | | | | | | | | | | | |
| 2798 ± 2 (n = 3) | 0.26 ± 0.11 | 43.4 ± 18.0 | 8.0 ± 0.0 | 4.6 ± 3.2 | 0.35 ± 0.17 | 24.0 ± 16.3 | 221.7 ± 138.7 | 0.74 ± 0.29 | 0.46 ± 0.40 | 7.1 ± 1.1 | 0.59 ± 0.74 | 3.5 ± 0.1 | 26.2 ± 9.8 |
| *Z. mays, P. vulgaris* | | | | | | | | | | | | | |
| 3007 ± 8 (n = 3) | 0.33 ± 0.11 | 40.8 ± 5.8 | 7.9 ± 0.1 | 4.5 ± 3.2 | 0.30 ± 0.15 | 26.4 ± 9.4 | 327.3 ± 220.8 | 1.23 ± 0.66 | 0.51 ± 0.36 | 18.4 ± 14.3 | 1.14 ± 1.21 | 3.5 ± 0.5 | 19.2 ± 5.6 |
| *Z. mays, V. faba, L. usitatissimum* | | | | | | | | | | | | | |

EC—electrical conductivity, CaCO$_3$—calcium carbonate, pH—hydrogen potential, OM—organic matter, N—nitrogen, P—phosphorus, K—potassium, B—boron, Cu—copper, Fe—iron, Mn—manganese, Zn—zinc, CEC—cation exchange capacity (cations: Ca$^{2+}$, Mg$^{2+}$, K$^+$ and Na$^+$), n- experimental plot *—standard deviation (±).

### 3.2. Soil Properties of Tara Forests

In Table 2, the location of the plots and the texture of the soils are shown. The soils of natural environments are primarily clay sandy loam, whereas the soils of the agroforestry environment are mostly clay loam. In general, the soils of the experimental plots evaluated in the present research have alkaline pH, high levels of K, OM, and CEC, an average

concentration of Mn and Cu, medium to high levels of B, Fe, and Zn; and low to medium P (Table 3).

In PCAs, the soil variables with the most significant contribution to altitude classes and environments were sand, N, OM, and CEC (Figure 2). In the altitude classes, it is observed that there is no pattern of variability as a function of altitude; however, they are randomly discriminated, showing spatial variability at altitude 2388 m associated with higher sand content and at altitude 2798 m associated with higher content of N, OM, and CEC (Figure 2a). However, in the agroforestry environment, soil properties remain similar to those in the natural environment (Figure 2b).

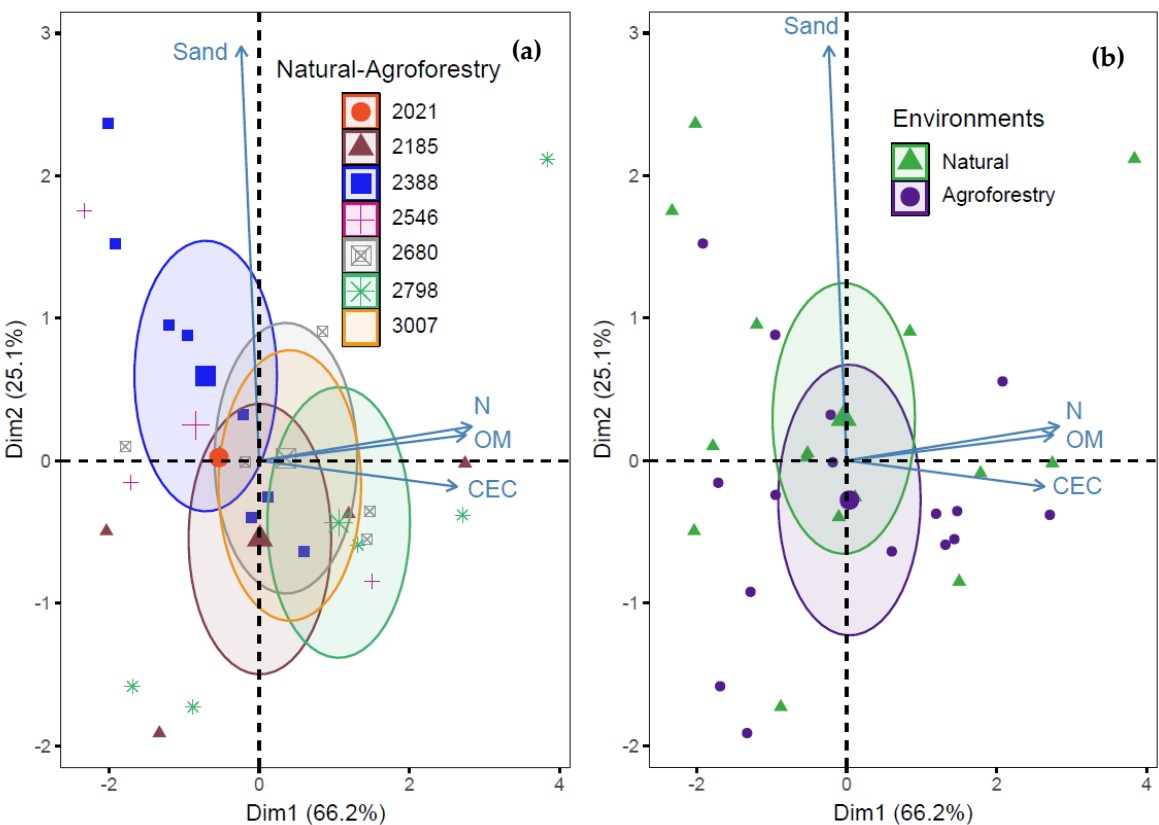

**Figure 2.** Principal component analysis (PCA) of organic matter (OM), cation exchange capacity (CEC), sand, and nitrogen (N) in the soil. (**a**) All altitude classes within the natural and agroforestry environments. (**b**) All plots in the natural and agroforestry environments, Cajamarca region, Andes Mountain range, Peru.

### 3.3. Tara Dendrometry in a Natural and Agroforestry System

Table 4 shows the tree height, crown height, crown diameter, stem diameter, and stem number at altitudes for the natural and agroforestry environment that are grouped and individually.

At altitudes (Figure 3a), the PCA for the natural and agroforestry environments grouped showed variation at altitude 3007 m, with greater stem diameter compared to altitudes of 2388 and 2798 m. In the individual PCAs for the different natural and agroforestry environments, there were no variations among altitudes (Figure 3c) in the natural environment, despite having presented significant correlations of the dendrometric variables of the individuals of Tara with the altitude, temperature, and physical and chemical properties of the soil (Figure S7a). On the other hand, in the agroforestry environment, it presented more significant spatial variation among altitudes (Figure 3b), and also possessed significant correlations of the dendrometric variables of the individuals of Tara with altitude, temperature, and physical and chemical properties of the soil (Figure S7b).

**Table 4.** Dendrometric variables of Tara trees in altitude class and natural and agroforestry environments.

| Altitude-Environment * | Density | Tree Height | Crown Height | Crown Diameter | Stem Diameter | Stem Number |
|---|---|---|---|---|---|---|
| | pl/200 m² | m | | | cm | |
| Natural—Agroforestry (n′ = 287) | 9.3 ± 8.4 | 5.3 ± 1.7 | 3.0 ± 1.3 | 4.1 ± 1.8 | 13.0 ± 9.4 | 3.2 ± 3.4 |
| 2021 (n = 1, n′ = 10) | 10.0 | 5.3 ± 1.5 | 3.5 ± 1.0 | 3.7 ± 1.2 | 11.8 ± 6.0 | 2.9 ± 2.3 |
| 2185 ± 3 (n = 4, n′ = 72) | 14.4 ± 10.4 | 6.6 ± 1.3 | 3.3 ± 1.3 | 4.4 ± 1.9 | 13.1 ± 7.7 | 3.4 ± 3.8 |
| 2388 ± 4 (n = 8, n′ = 102) | 12.8 ± 9.5 | 5.3 ± 1.9 | 3.0 ± 1.3 | 3.6 ± 1.5 | 9.9 ± 6.0 | 4.2 ± 3.9 |
| 2546 ± 6 (n = 3, n′ = 41) | 13.7 ± 11.0 | 4.6 ± 1.4 | 2.8 ± 1.3 | 4.3 ± 1.6 | 13.8 ± 9.3 | 2.5 ± 2.7 |
| 2680 ± 7 (n = 5, n′ = 26) | 5.2 ± 2.8 | 5.1 ± 1.1 | 3.3 ± 0.8 | 5.4 ± 1.9 | 18.0 ± 8.3 | 2.1 ± 1.2 |
| 2798 ± 2 (n = 5, n′ = 29) | 5.8 ± 5.5 | 3.7 ± 0.9 | 2.0 ± 0.9 | 3.5 ± 2.0 | 15.1 ± 16.3 | 2.2 ± 2.2 |
| 3007 ± 7 (n = 4, n′ = 7) | 1.8 ± 1.0 | 4.9 ± 1.7 | 3.4 ± 1.5 | 5.7 ± 2.3 | 28.4 ± 13.3 | 2.0 ± 1.5 |
| Natural (n′ = 226) | 15.1 ± 8.8 | 5.3 ± 1.7 | 2.8 ± 1.2 | 3.8 ± 1.5 | 12.0 ± 8.6 | 3.10 ± 3.22 |
| 2021 (n = 1, n′ = 10) | 10.0 | 5.3 ± 1.5 | 3.5 ± 1.0 | 3.7 ± 1.2 | 11.8 ± 6.0 | 2.9 ± 2.3 |
| 2185 ± 4 (n = 3, n′ = 64) | 21.3 ± 6.1 | 6.5 ± 1.2 | 3.1 ± 1.2 | 4.0 ± 1.6 | 12.1 ± 6.1 | 3.0 ± 3.4 |
| 2388 ± 5 (n = 4, n′ = 78) | 19.5 ± 9.3 | 5.2 ± 1.9 | 2.8 ± 1.3 | 3.4 ± 1.2 | 9.6 ± 5.9 | 3.9 ± 3.8 |
| 2546 ± 6 (n = 2, n′ = 36) | 18.0 ± 11.3 | 4.6 ± 1.5 | 2.7 ± 1.3 | 4.3 ± 1.5 | 13.0 ± 9.0 | 2.7 ± 2.8 |
| 2680 ± 8 (n = 2, n′ = 13) | 6.5 ± 5.0 | 4.5 ± 0.6 | 2.7 ± 0.4 | 4.7 ± 1.4 | 18.0 ± 7.3 | 1.9 ± 1.3 |
| 2798 ± 2 (n = 2, n′ = 23) | 11.5 ± 2.1 | 3.9 ± 0.9 | 2.1 ± 0.9 | 3.5 ± 2.0 | 14.9 ± 17.3 | 2.1 ± 1.9 |
| 3007 (n = 1, n′ = 2) | 2.0 | 3.4 ± 1.3 | 1.8 ± 0.8 | 3.6 ± 0.4 | 16.6 ± 1.4 | 1.0 ± 0.0 |
| Agroforestry (n′ = 61) | 3.8 ± 2.1 | 5.4 ± 1.8 | 3.7 ± 1.4 | 5.26 ± 2.25 | 16.8 ± 11.3 | 3.8 ± 3.8 |
| 2185 ± 4 (n = 2, n′ = 8)<br>*M. sativa, L. multiflorum* | 4.0 ± 0.0 | 7.2 ± 1.7 | 4.9 ± 1.4 | 7.2 ± 1.7 | 21.2 ± 13.6 | 6.1 ± 5.7 |
| 2388 ± 3 (n = 4, n′ = 24)<br>*Z. mays, P. vulgaris, T. aestivum* | 6.0 ± 2.0 | 5.5 ± 1.8 | 3.7 ± 1.4 | 4.4 ± 2.0 | 10.9 ± 6.3 | 4.9 ± 4.3 |
| 2546 (n = 1, n′ = 5)<br>*Z. mays, P. vulgaris* | 5.0 | 4.4 ± 1.4 | 3.2 ± 1.4 | 4.9 ± 2.0 | 19.4 ± 10.7 | 1.6 ± 0.9 |
| 2680 ± 5 (n = 3, n′ = 13)<br>*Z. mays, P. vulgaris, M. sativa, L. multiflorum* | 4.3 ± 0.6 | 5.6 ± 1.2 | 3.8 ± 0.8 | 6.1 ± 2.2 | 18.1 ± 9.6 | 2.2 ± 1.2 |
| 2798 ± 2 (n = 3, n′ = 6)<br>*Z. mays, P. vulgaris* | 2.0 ± 1.7 | 3.3 ± 0.9 | 1.7 ± 1.0 | 3.6 ± 2.0 | 15.9 ± 13.1 | 2.5 ± 3.2 |
| 3007 ± 8 (n = 3, n′ = 5)<br>*Z. mays, V. faba, L. usitatissimum* | 1.7 ± 1.2 | 5.6 ± 1.5 | 4.0 ± 1.1 | 6.6 ± 2.2 | 33.1 ± 13.0 | 2.4 ± 1.7 |

*- standard deviation (±), n—experimental plot, n′—Tara plants.

In the agroforestry environment (Figure 3b), the altitude of 2185 m presented higher values of crown diameter, tree height, and crown height compared to those at 2388, 2546, and 2798 m altitudes. Altitudes 2388 and 2546 m were associated with seasonal crops of *Z. mays*, *P. vulgaris*, and *T. aestivum*, while altitude 2185 m was associated with perennial pasture *M. sativa* and *L. multiflorum* (Tables 2–4). Conversely, the altitude of 2798 m, in an agroforestry environment, is associated with the lowest values of crown height, total height, and crown diameter when compared to altitudes of 2185, 2388, and 2680 m (Figure 3b). The low levels of Fe at altitude 2798 m (Figure S2) may affect the development of Tara individuals; in addition, Fe has a negative correlation with $CaCO_3$ and pH (r = −0.60; $p < 0.001$ and r = −0.67 $p < 0.001$), respectively (Figure S7b).

The altitude of 2388 m in the agroforestry environment (Figure 3b) presented a higher number of stems compared to other altitudes. Regarding soil properties, the altitude of 2388 m has lower levels of $CaCO_3$ (Figure S2). Likewise, in the agroforestry environment, sites at 3007 m (Figure 3b) possess the largest stem diameter compared to the altitudes of 2185, 2388, 2546, and 2798 m. Furthermore, $CaCO_3$ is positively correlated with the altitude (r = 0.28, $p < 0.05$) and with stem diameter (r = 0.49, $p < 0.001$) of the Tara tree (Figure S7b).

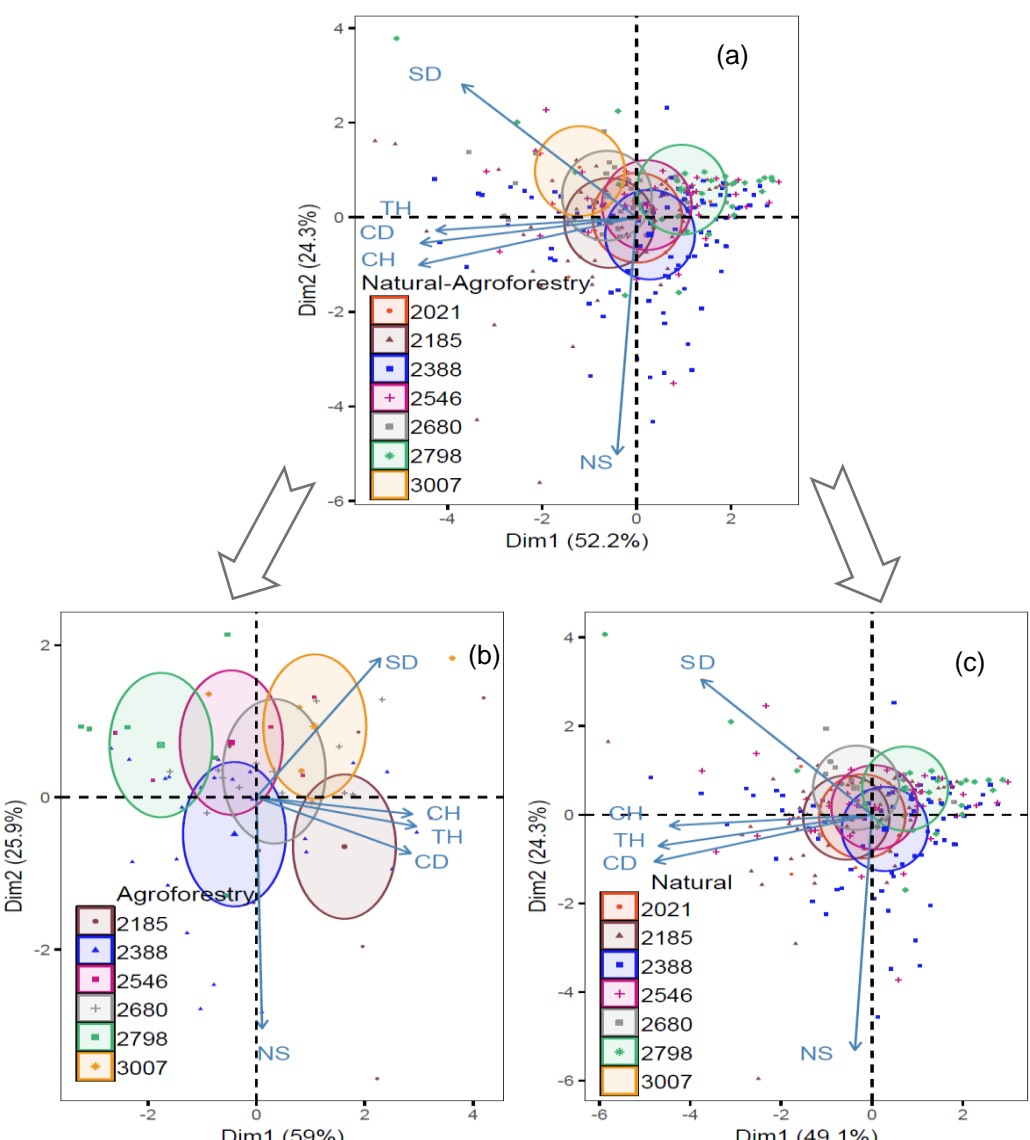

**Figure 3.** Principal component analysis (PCA) of number of stems (NS), crown diameter (CD), tree height (TH), crown height (CH), and stem diameter (SD) of Tara. Natural and agroforestry environments (**a**), altitudes for agroforestry environments (**b**), and altitudes for natural environments (**c**).

In the altitude classes for the agroforestry environment (Figure 4), the dendrometric variables of the Tara tree did not show differences among altitudes by the bootstrap method, except for tree height, which showed significant differences (CI: 95%) with a higher value at altitude 2185 m compared to altitude 2798 m (Figure 4). This difference corroborates the PCA (Figure 3b). In addition, the altitude of 2798 m has a low level of Fe (Figure S2).

In PCAs for natural and agroforestry environments grouped (Figure 5a), there was no variation in the dendrometric variables of Tara. However, PCAs performed individually for altitudes of 2185 m (Figure 5b), 2680 m (Figure 5e), and 3007 m (Figure 5g) indicate differences between natural and agroforestry environments, with higher values in some stems, crown diameter, tree height, crown height, and stem diameter for the agroforestry environment, except for the number of stems at altitude 2680 m, which are not different. At 2185 m altitude, the density of individuals of Tara was lower (Figure S3) for the agroforestry environment; in the soil, the contents of Zn and Mn were also lower, while the contents of Cu and clay were higher for the agroforestry environment (Figure S4). Similarly, at an altitude of 2680 m, the agroforestry environment has higher Mn, silt, and clay values in the

soil (Figure S5). At an altitude of 3007 m, the agroforestry environment has higher levels of CaCO$_3$, P, K, and B, but with lower levels of Fe in the soil (Figure S6).

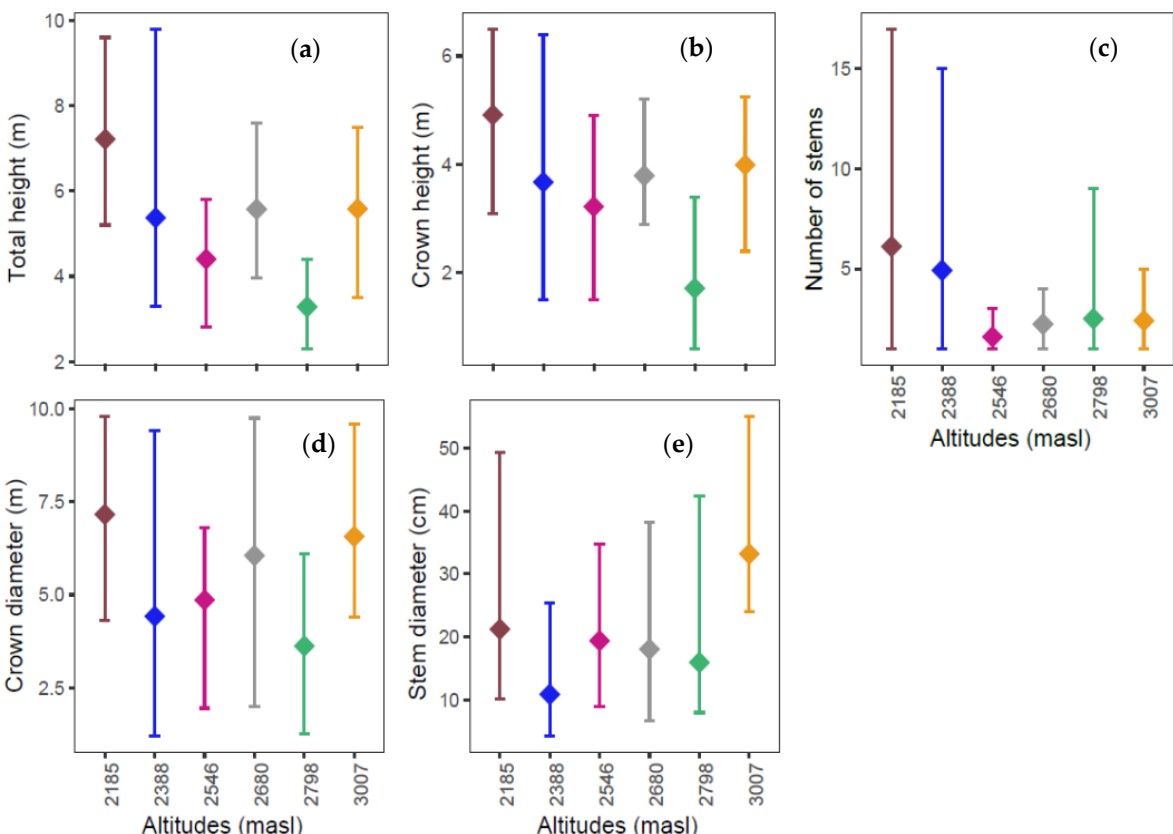

**Figure 4.** Dendrometry of the Tara tree in an agroforestry environment; tree height (**a**), crown height (**b**), number of stems (**c**), crown diameter (**d**), stem diameter (**e**). The mean was followed by the bootstrap method's non-parametric confidence interval (CI: 95%).

The PCA showed spatial differences at 2185 m (Figure 5b), 2680 m (Figure 5e), and 3007 m (Figure 5g), but by the bootstrap method (Figure 6), it only showed significant differences at altitude 3007 m (Figure 6c) with higher values of crown height, crown diameter, and stem diameter of the Tara tree for the agroforestry environment. At an altitude of 3007 m, the soil had higher CEC and Fe for the natural environment and the agroforestry environment, and it had higher levels of CaCO$_3$, P, K, and B in the soil (Figure S6).

### 3.4. Tree Temperature-Altitude-Soil-Dendrometry Interaction

In correlations of altitude, temperature, soil properties, and dendrometric variables of the tree, the natural environment presents more significant correlations, with higher coefficient values (Figure S7a) than the agroforestry environment (Figure S7b). The natural environment had weak correlations between altitude and tree height (r = −0.38). Soil properties in the natural environment possessed moderate correlations of altitude with Mn (r = −0.53), CEC (r = 0.42), and Fe (r = −0.47), and weak correlations with silt (r = −0.38), P (r = −0.38); similarly, these properties correlate with temperature.

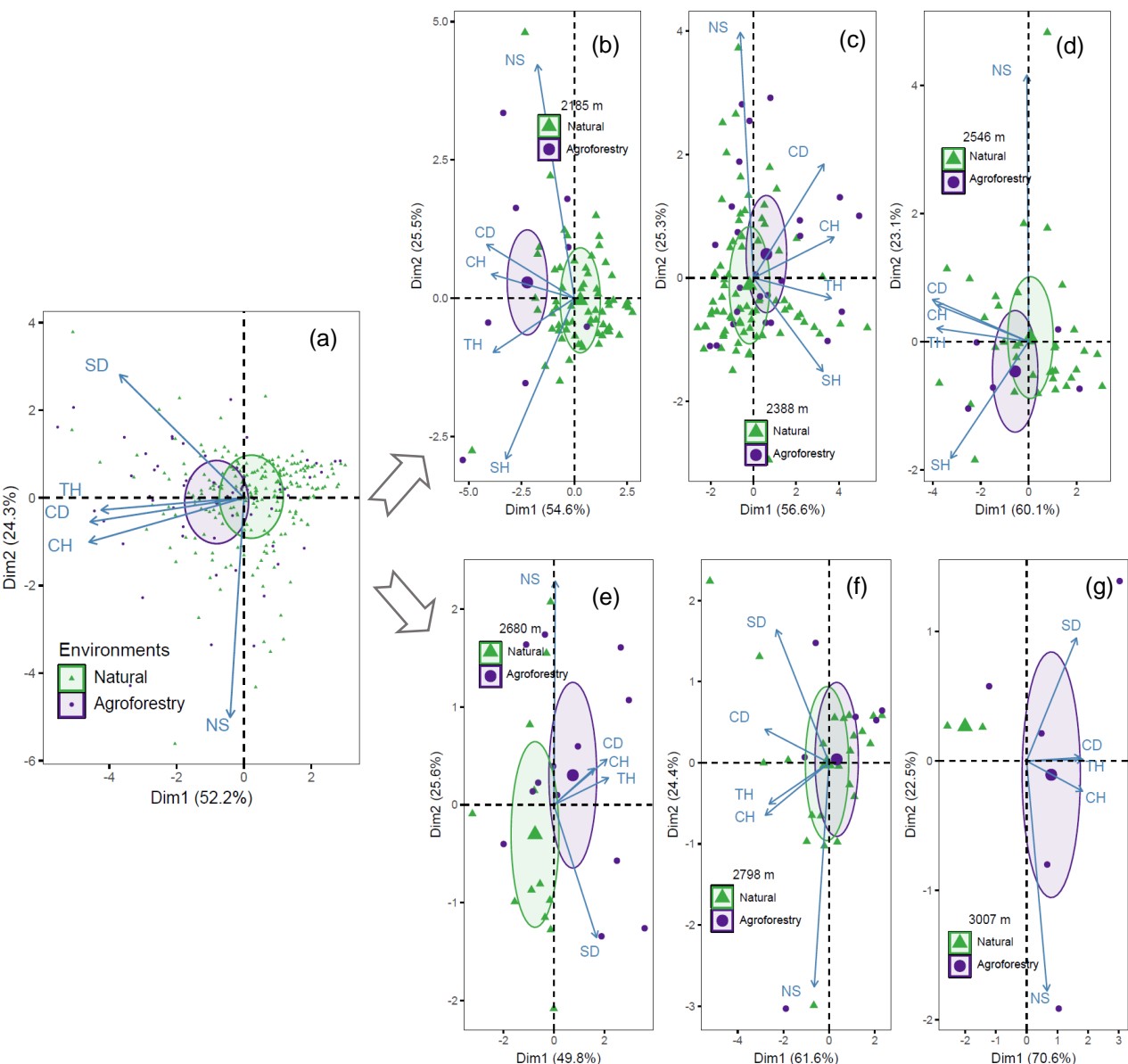

**Figure 5.** Principal component analysis (PCA) of the number of stems (NS), crown diameter (CD), tree height (TH), crown height (CH), and stem diameter (SD) of Tara. All plots (**a**), altitude 2185 m (**b**), altitude 2388 m (**c**), altitude 2546 m (**d**), altitude 2680 m (**e**), altitude 2798 m (**f**), and altitude 3007 m (**g**); for natural and agroforestry environments.

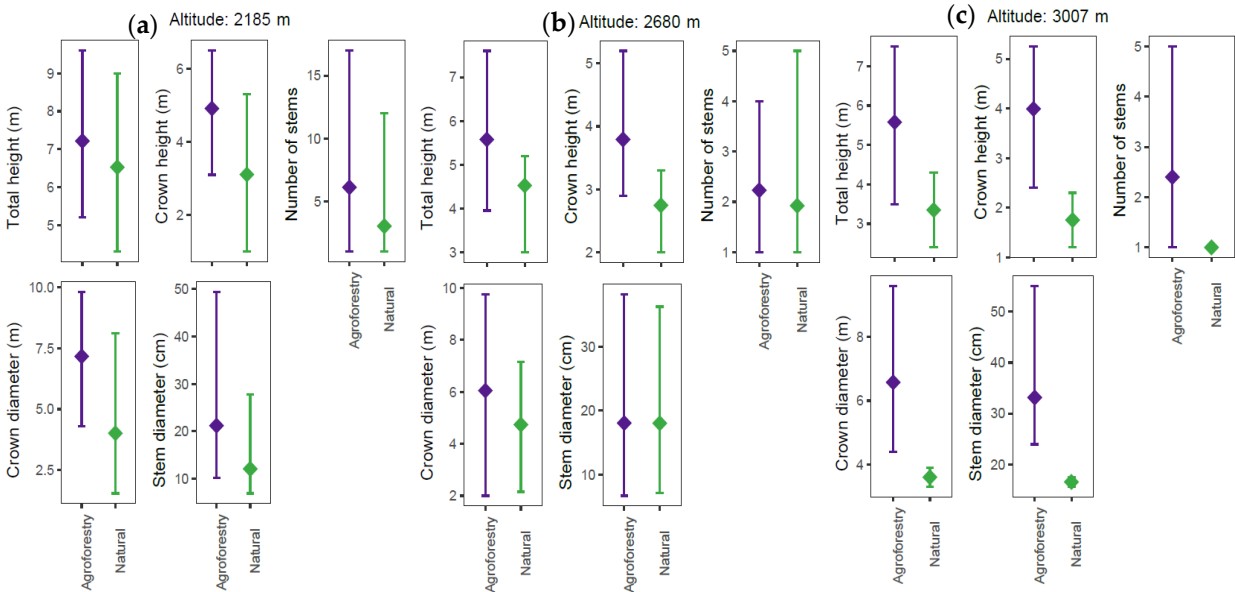

**Figure 6.** Dendrometry of the Tara tree: tree height, crown height, number of stems, crown diameter, and stem diameter. Altitude 2185 m (**a**), altitude 2680 m (**b**), and altitude 3007 m (**c**) for natural and agroforestry environments. The mean was followed by the bootstrap method's non-parametric confidence interval (CI: 95%).

## 4. Discussion

### 4.1. Temperature and Precipitation

The minimum temperature of the altitudinal gradient was $13.4 \pm 0.4$ °C in the altitude class of 3007 m ($3007.8 \pm 72.0$ m) (Table 2); probably, this temperature and/or altitude approaches the abiotic limit of Tara because, in the study area above this altitude, no Tara trees were found either as natural remnants or integrated into agroforestry systems. The altitudinal gradients in the tropical and subtropical forests of the Andes generate biotic (mutualisms/antagonisms) and abiotic (temperature, rainfall, and soils) limits, producing clear vegetation differentiation [31,32]. In other regions of Peru, Balaguer et al. [33] reported natural forests of Tara at 1600 m in tropical pre-mountain (Huánuco) and at 2200, 2400, and 2700 m in subtropical low mountain (Ayacucho and Apurímac) areas. In Ayacucho, De La Cruz-Arango et al. [34], in turn, found Tara trees between 1550 and 3000 m altitude.

### 4.2. Soil Properties of Natural Environments of Tara

Soils associated with Tara are alkaline, with high OM, K, and CEC content, and medium to low contents in the other properties evaluated (Table 3). In forests of Tara introduced in the Peruvian desert, Balaguer et al. [33] found soils with high levels of K (280 mg dm$^{-3}$), OM (3.2%), P (47.1 mg dm$^{-3}$), Fe (31.9 mg dm$^{-3}$), and Mn (75.9 mg dm$^{-3}$); but with a strongly acidic pH (4.9). This pH was the opposite of the results found in the current research, in which the pH ranged from 7.7 to $8.0 \pm 0.4$, suggesting that Tara has wide adaptability to soil pH.

There were no differences among environments when all plots were analyzed for N, OM, CEC, and soil sand (Figure 2b). This suggests that there were no changes in the soil due to agroforestry use, keeping soil properties similar to natural conditions. These results show that agroforestry system soils are as efficient as natural systems, maintaining properties along the altitudinal gradient. This occurred possibly because along the altitudinal gradient, there was a high content of OM and CEC, which attenuates changes in soil properties, leading to similarity between natural and agroforestry environments. Soils with CEC above 10 cmolc.kg$^{-1}$ generally have high buffering capacity [35]. Sylvester et al. [36] state that the lack of significant differences in soil properties in the Peruvian tropical Andes, in

environments with natural forests and pasture, is due to OM, which maintains soil pH and cation contents.

In the altitude classes for the natural and agroforestry environment (Figure 2a), PCA in soil properties at altitude 2798 m is associated with higher OM, CEC, and N values, and at altitude 2388 m with higher sand content. According to Hassink et al. [37], soils with higher sand content produce significantly greater N mineralization from the OM, in contrast to weak-clay and clayey soils in which fine porosity protects OM and N. In the Peruvian tropical Andes, soils are stony (40 to 80%), with varying depths (20 to 70 cm) depending on the topography [36]. Temperature has an influence on the physical and chemical properties of the soil [38], since edaphic properties depend on the parent material, forming Cambisols and Umbrisols [21], without showing clear trends regarding the content of N and P [21,24]). In tropical forests with an altitude gradient, the physical and chemical properties of the soil do not vary between natural and agroforestry systems but present higher values in terms of fertility when compared to soils exclusively used for pastures. In addition, concerning soils of natural mountain forests, [39] determined, through PCA, that the variables with the most significant contribution were the cations $Mg^{2+}$, $Ca^{2+}$, $K^+$, pH, OM, N, and P. Similarly, Slik et al. [40] observed that the variables with the most outstanding contribution to PCA were CEC, N, pH, soil texture, and organic carbon. Likewise, in the Ecuadorian Andes, Unger et al. [41] established that N and soil cations ($Ca^{2+}$, $Mg^{2+}$ and $K^+$) lacked significant differences over a range of 1500 m in altitude. The soils of the tropical Andes are heterogeneous with high OM contents [18]; however, the ratio of C/N increases between 1000 and 3000 m of elevation, generating a low OM mineralization and consequently, poor nutrient availability [38].

Tara is not demanding in terms of soil quality; it is found in stony, degraded, and even lateritic soils with low productivity; however, when in loam and sandy loam soils, pod productivity is good [10].

### 4.3. Dendrometric Variables of Tara in Natural and Agroforestry Systems

Agroforestry systems, associated for a long time (>10 years) with perennial pastures of *M. sativa*; a plant well-known for its biological fixative N capacity and deep roots that provide nutrients such as P and K; favor the development of the Tara tree at altitude 2185 m, differently from the soil with low Fe contents at altitude 2798 m (Figure 3b; Figure S2; Tables 2–4). In the soil of the agroforestry environment of *Morus* spp. Associated with the pasture of *M. sativa*, there was greater availability of N, P, and K from bacterial activity, in contrast to the soil of the forest environment containing *Morus* spp. only [42].

On the other hand, Fe is essential for electron transport in cell mitochondria in plants' photosynthesis [43]. In fruit-bearing species, Fe deficiency produces chlorosis, with reduced growth and low production, especially in calcareous soils [44,45]. In alkaline soils, Fe is difficult to absorb by plants since it is found in poorly soluble oxides [43]. In this research, the soils generally are calcareous, with a strongly alkaline pH (Table 1, Figure S2). Low Fe availability in calcareous soils negatively affects *Sorghum bicolor* plants [46].

In addition, in the high-altitude classes (Figure 3), an altitude of 3007 m in the PCA has a larger stem diameter in the agroforestry environment (Figure 3b), which is associated with higher levels of $CaCO_3$ in the soil (Figure S2). In *Acer saccharum* and *Fagus grandifolia* forests, higher $CaCO_3$ favored larger stem diameter of these species [47].

Comparing the natural and agroforestry environments in general (Figure 5a) and individual analysis, at altitudes 2388 m (Figure 5c), 2546 m (Figure 5d), and 2798 m (Figure 5f), there is no difference among environments; Tara individuals show plasticity under these conditions in the natural environment, and these plants have greater intra- and interspecific competition. Dendrometric variables are similar to those individuals in agroforestry environments at these altitudes.

In natural systems, Tara individuals are emergent trees; present management is tasked with reduction of the shrub and herbaceous strata vegetation to facilitate the harvest of the pods in production. Tara, in natural environments, is found primarily in steep areas

with densities of up to 1400 ind.ha$^{-1}$. Tara individuals in agroforestry environments are found in densities of up to 400 ind.ha$^{-1}$, so that the availability of light, water, and nutrients in the soil allows economically viable productions, both for crops and for Tara. In agroforestry environments, crops are rotated annually, aiming to increase the productivity of the environment with crops of high nutritional and commercial value, such as *Solanum tuberosum* (Pureja group), *Chenopodium quinoa*, *Smallanthus sonchifolius*, *Z. mays* (black and chulpe), *Amaranthus caudatus*, *Physalis peruviana*, and are perennially associated with fruit trees such as *Annona cherimolia* and *Persea americana* whose flowering periods do not match with Tara's flowering time, producing less competition for pollinators; the fruit trees complement each other and increase the presence of bees.

In the natural environments of the tropical forests of the Andes, Tara is found in the dominant stratum. When there is a change in land use, Tara becomes the primary option for agroforestry systems due to the production of pods and the canopy structure. This results in a lower density of leaves, which generates diffuse shade, enhancing acceptable production of associated crops. The differences between natural and agroforestry environments at altitudes 2185 m (Figure 5b), 2680 m (Figure 5e), and 3007 m (Figure 5g); with higher dendrometric variables of the tree in an agroforestry environment; may be subject to agricultural practices and crops associated with biological N fixation, such as *P. vulgaris*, *V. faba*, and *M. sativa*. In addition, organic fertilization and the method of soil preparation offer greater aeration and nutrients; eliminating vegetation before sowing crops reduces interspecific competition for water and nutrients.

Differences between environments at altitude 2185 m (Figure 5b) may be related to the higher density of individuals in the natural environment. This may be related to higher levels of Zn and Mn in the soil compared to the agroforestry environment. Agroforestry has higher levels of Cu and clay in the soil. Trees in the natural environment are probably affected by significant variable contents of Zn, Mn, and Cu in the soil and by the higher density of individuals (Figures S3 and S4). In orange trees, doses of 5 g.pL$^{-1}$ of Zn and 3.5 g.pL$^{-1}$ of Mn showed toxicity in sandy soils, affecting photosynthesis, and consequently, tree development. With these same doses, the effects were not noticeable in clayey soils; doses of 0.1 g.pL$^{-1}$ of Zn and 0.7 g.pL$^{-1}$ of Mn were sufficient as fertilizers for this fruit tree [48]. The higher density of individuals in the natural environment reduces the space between plants, producing greater intraspecific competition, which affects the development of the tree. The greater space between plants in an agroforestry environment can favor horizontal and vertical growth.

Similar to the altitude of 2680 m (Figure 5e), the difference between environments is probably due to the natural environment containing higher values of sand, taking into account the observation that sand soils have a negative correlation with the CEC (r = −0.23, *p* < 0.001) (Figure S7b). Clay loam soil favors CEC, as it presents a positive correlation (r = 0.44, *p* < 0.001) between clay and CEC in the agroforestry environment (Figures S5 and S7a).

Differentiation of the tree's dendrometric variables among environments at an altitude of 3007 m (Figure 5g) was possible because the agroforestry environment presented higher levels of K, P, B, and CaCO$_3$; and lower Fe content in the soil compared to the natural environment (Figure S6). Likely, Fe did not occur in low amounts, and K, P, B, and CaCO$_3$ favored the development of the Tara tree in an agroforestry environment. This suggests higher soil nutrient contents in the agroforestry environment, mainly K, P, B, and CaCO$_3$.

## 5. Conclusions

The anthropic effect on the environment did not significantly alter the soil's CEC, OM, N, and sand. No dendrometric variables varied as a function of altitude and temperature in PCAs run in a natural environment. However, the agroforestry environment presented a greater crown diameter, tree height, and crown in the altitude class 2185 m associated with *M. sativa* and *L. multiflorum.* The opposite behavior was found in these variables in the altitude class 2798 m associated with low Fe content. In the 3007 m altitude sites, larger

stem diameter was associated with higher levels of CaCO$_3$. Tara's crown diameter, crown height, and tree height were significantly higher in agroforestry environments only at 3007 m of altitude. Variations in soil nutrients, mainly Fe and CaCO$_3$, suggest interference in the development of Tara individuals in natural and agroforestry environments. It is important to carry out further studies for a better understanding of the relationship among the production of Tara pods and soil fertility in altitudinal variation, aiming to improve the income and employment of family farmers who exploit Tara in the Peruvian Andes.

**Supplementary Materials:** The following supporting information can be downloaded at: https://www.mdpi.com/article/10.3390/agronomy13020282/s1.

**Author Contributions:** Conceptualization, H.M.-O.; methodology, H.M.-O., R.S.S.A. and F.D.A.L.; software, H.M.-O.; validation, F.W.C.-K., P.M.S. and G.L.V.; formal analysis, R.S.S.A., F.D.A.L. and G.L.V.; investigation, H.M.-O., C.A.-R., L.F.S.D., G.S., E.N.B. and R.M.B.-L.; resources, R.M.B.-L.; data curation, H.M.-O., R.S.S.A. and F.D.A.L.; writing—original draft preparation, H.M.-O.; writing—review and editing, H.M.-O., F.W.C.-K. and P.M.S.; visualization, H.M.-O. and F.W.C.-K.; supervision, R.S.S.A. and F.D.A.L.; project administration, R.M.B.-L.; funding acquisition, F.D.A.L. All authors have read and agreed to the published version of the manuscript.

**Funding:** The first author thanks to the Coordenação de Aperfeiçoamento de Pessoal de Nível Superior—Brasil (CAPES)—Finance Code 001, for funding the research, through the Program of Alliances for Education and Training (PART) and the Organization of American States (OAS).

**Institutional Review Board Statement:** Not applicable.

**Informed Consent Statement:** Not applicable.

**Data Availability Statement:** Not applicable.

**Conflicts of Interest:** The authors declare no conflict of interest.

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
