# Peer review of "Tara (Caesalpinia spinosa) in Natural and Agroforestry Systems under an Altitudinal Gradient in the Peruvian Andes: Responses to Soil and Climate Variation"

_agronomy, doi:10.3390/agronomy13020282_

Round 1

Reviewer 1 Report

Attached is the manuscript with some suggestions inserted

Reviewer 2 Report

General comments

It is an interesting manuscript. However, the authors need to improve some details.

The authors must change the manuscript format because they did not use the agronomic journal template (MDPI).

The authors should change the references' style because it does not follow the agronomic journal (MDPI).

Specific comments

What is the idea of measuring the concentration of B, Zn, Cu, and Mn in the soils? They are micronutrients, so their effects on soil properties are very low.

Line 12: Why did you use the K element since it is inside the CEC?

Line 21: You must use content after N.

Line 38: You should change today by currently.

Line 60: You should delete according because you have used it in the previous line.

Line 104: You should indicate if you have used five kg or one kg soil subsamples.

Line 104: Why did you use a depth of 0 to 40 cm?

Line 108: Why did you dry the soil samples at 17.7 °C?

Line 109: You should put 0 °C instead of 0.0 °C.

Line 113: You must indicate if CaCO3 is limestone, Calcium, or calcium carbonate.

Line 114: What is a standard methodology? You must add a reference.

Line 188: Why did you indicate that high elements were found? You need to compare it with other soil.

Line 193: Why did you indicate that higher content of N, OM, and CEC? You need to compare it with other soils.

Line 203: You can not mention Figure 7Sa if you did not mention Figure S2 first.

Line 268: You should check out the word mg dm-3. Use units of mg kg-1 or µg kg-1.

Line 291: You have to revise the sentence.

Line 294: Divalent cations are written as Mg2+, Ca2+

Line 312-317. What is the importance of these sentences for your research?

Line 351-352: You have to Improve the idea.

Line 255. You must use mg kg-1 or µg kg-1.

Line 373:  CaCO3 is a nutrient?

Line 376. You must indicate that it is the sand fraction.
